# Dissipative quantum North-East-Center model: steady-state phase diagram, universality and nonergodic dynamics

**Pietro Brighi[1⋆] and Alberto Biella[2,3†]**

**1** Faculty of Physics, University of Vienna, Boltzmanngasse 5, 1090 Vienna, Austria
**2** Pitaevskii BEC Center, CNR-INO and Dipartimento di Fisica, Università di Trento, I-38123 Trento, Italy
**3** INFN-TIFPA, Trento Institute for Fundamental Physics and Applications, I-38123 Trento, Italy

⋆ pietro.brighi@univie.ac.at ,    † alberto.biella@cnr.it

## Abstract

In this work we study the dissipative quantum North-East-Center (NEC) model: a two-dimensional spin-1/2 lattice subject to chiral, kinetically constrained dissipation and coherent quantum interactions. This model combines kinetic constraints and chirality at the dissipative level, implementing local incoherent spin flips conditioned by an asymmetric majority-vote rule. Through a cluster mean-field approach, we determine the steady-state phase diagram of the NEC model under different Hamiltonians, consistently revealing the emergence of two distinct phases, bistable and normal, across all cases considered. We further investigate the stability of the steady-state with respect to inhomogeneous fluctuations in both phases, showing the emergence of instabilities at finite wavevectors in the proximity of the phase transition. Next, we study the nonergodicity of the model in the bistable phase. We characterize the dynamics of minority islands of spins surrounded by a large background of spins pointing in the opposite direction. We show that in the bistable phase, the minority islands are always reabsorbed by the surrounding at a constant velocity, irrespectively of their size. Finally, we propose and numerically benchmark an equation of motion for the reabsorption velocity of the islands where thermal and quantum fluctuations act independently at linear order.

# 1 Introduction

The development of experimental platforms for quantum simulation has stimulated the study of novel non-equilibrium phenomena, arising in exotic systems [1–4]. In this context, quantum kinetically constrained models (KCMs) have emerged as a fascinating framework for the study of non-equilibrium physics [5–7]. First introduced in the context of classical glasses [8], KCMs are systems whose dynamics have to satisfy a certain set of rules, giving rise to non-ergodic behaviors and slow relaxation. In the quantum realm, KCMs have shown a plethora of interesting phenomena, ranging from anomalous dynamical properties [1, 9–12] to weak ergodicity breaking [13–15] and Hilbert space fragmentation [16–18].

The interplay of kinetic constraints and dissipation has been studied both in the context of stability of the nonergodic features of KCMs when coupled to external environments [19,20], and in the context of kinetically constrained dissipative processes [21–26]. The latter, in particular, have shown that the interplay of dissipative kinetic constraints and coherent dynamics can lead to active phases of matter [27] as well as critical [28,29] and heterogeneous [24–26] dynamics.

The intersection between these two classes of systems (KCMs and open quantum systems) opens an exciting playground for non-equilibrium physics. Indeed, in open quantum many-body systems, interactions and dissipation compete and non-equilibrium phases of matter emerge [30] triggering dissipative criticalities [31–34] and stabilizing phases forbidden at thermal equilibrium [35–37].

A promising avenue in the study of dissipative kinetically constrained systems corresponds to chiral KCMs, where constraints are spatially asymmetric. These have garnered significant interest in isolated systems, due to their peculiar dynamics [7, 11, 12, 38, 39], whereas their dissipative counterpart remains relatively unexplored. Particularly, two-dimensional lattice models have escaped thorough investigation due to the limitations in theoretical and numerical approaches. However, their study opens new directions in the interplay of dissipation and kinetic constraints due to the possibility of designing angular-dependent chiral processes and, at the same time, supporting disspative phase transitions. In this context, the North-East-Center (NEC) model provides a fascinating example of a two-dimensional dissipative KCM.

Introduced by Toom in 1974 [40, 41] as a classical majority vote model in the context of cellular automata, it generalizes the East model to a two-dimensional plaquette and serves as a toy model for the investigation of the effect of inversion symmetry breaking in 2-dimensional systems. It was later shown in Ref. [42] that the dynamics of the NEC model lead to a bistable

phase, where two different steady-states emerge even in presence of classical noise [43], i.e. when errors with respect to the majority-vote rule are introduced leading to an effective temperature. More recently, a variational analysis has shown that the bistability in the NEC model persists even in presence of quantum fluctuations [44], thus shaking the common belief that quantum many-body systems cannot host robust bistable phases. The variational approach exploited in Ref. [44] (originally proposed in Ref. [45]), however, does not allow to scale the number of physical sites included in the treatment and thus systematically analyze the effect of quantum fluctuations at increasing distances. This exact treatment of short-range physics has been shown to be crucial to determine the structure of the steady-state phase diagram (see, e.g., Ref. [46]) since it is known that mean-field-like decoupling can lead to artificial multiple stable solutions [47]. Establishing whether these findings are stable against the systematic inclusion of short-range quantum correlations and universal with respect to different classes of coherent Hamiltonain interactions, remain exciting open questions. Furthermore, the non-ergodic dynamics expected in the bistable region remains largely unexplored in the quantum scenario. To this aim it is necessary to access the real-time evolution of spin islands. Again, this aspect has not be addressed in [44] and calls for further investigations.

In this work, we compute the steady-state of the system and obtain the phase diagram for the relevant parameters in the model, highlighting a finite ferromagnetic bistable region. In particular, we study how different sources of quantum fluctuations, i.e. different Hamiltonians, affect bistability. Specifically, we investigate Hamiltonians without kinetic constraints, models with constraints that respect the NEC symmetry, and models with constraints that conflict with the NEC geometry. Interestingly, we find that bistability in the NEC model is a universal feature persisting irrespective of the microscopic details of quantum fluctuations. Nonetheless these details affect the extent of the bistable phase, and we show that the presence of kinetic constraints in the Hamiltonian results in strong bistability. We use a cluster mean-field (CMF) ansatz [46] to systematically include short-range correlations considering cluster of increasing size. We further introduce a inhomogeneous CMF method (iCMF) which allows to study the real time dynamics of large two-dimensional systems starting from non-translationally-invariant initial conditions.

We exploit the latter ansatz to investigate the dynamics in the bistable and normal phases of inhomogeneous initial states where an island of spins pointing in the same direction is surrounded by a background of spins in the opposite state. While in the normal phase the two regions quickly mix yielding a unique steady-state, in the bistable phase islands are always absorbed by their surrounding, irrespectively of their size. The study of dynamics then highlights how bistability is rooted in the nonergodic dynamics of the NEC model, where different initial states evolve to different steady-states with opposite magnetization. The absorption of *error islands* from a given background could be used as a strategy for quantum error correction [48–50], where an undesired local flip to the wrong state is always absorbed by the surroundings. We analyze the absorption velocity, showing that it is independent of island size in the bistable phase, and we propose a phenomenological picture for the velocity that includes the effect of both classical and quantum fluctuations.

The remainder of this paper is organized as follows. First, in Sec. 2 we introduce the jump operators and the Hamiltonians that specify the NEC model and the Lindblad master equation. In Sec. 3 we describe the CMF method including its inhomogeneous generalization used in the study of dynamics. In Sec. 4 we study in detail the phase diagram of the NEC model under the different sources of quantum fluctuations and in 5 we study the stability with respect to inhomogeneous perturbations. Finally, In Sec. 6, we study the absorption dynamics of the spin islands. In 7 we summarize our work, highlighting the potential of dissipative kinetically constrained models as an avenue for non-equilibrium phases of matter and suggesting the inhomogeneous cluster mean-field method as a viable way to study the dynamics of large

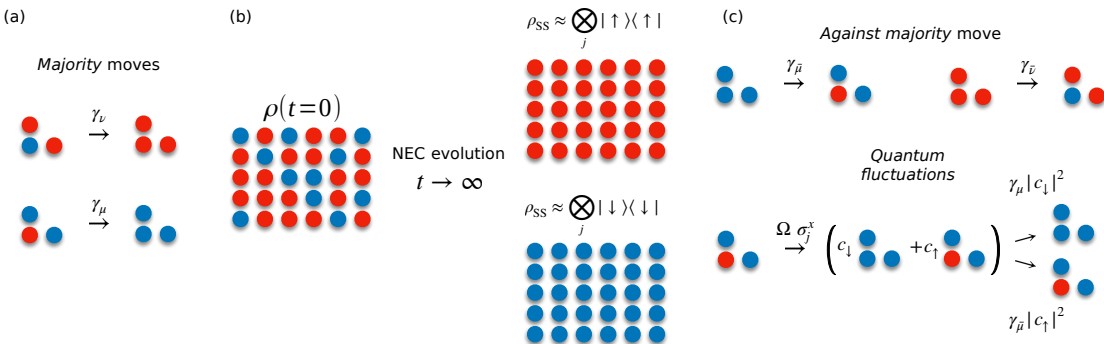

Figure 1: Pictorial representation of the various processes in the NEC model. (a): The majority moves implemented by the jump operators in Eq. (2,3) align the spin at the corner of the plaquette to the majority. (b): The chiral dissipative constraint imposed by the jump operators breaks ergodicity and leads to bistability. In the bistable region, the steady-state can have either a large positive or large negative magnetization, depending on the initial condition. Red and blue sites represent spin pointing upwards and downwards, respectively. (c): The rate of the majority moves can be lowered by classical fluctuations and coherent quantum dynamics. Classical fluctuations act as moves against the majority which decrease the effectiveness of the majority moves. In the sketch below the action of a local field on the Center site $\sigma_j^x$ with strength $\Omega$ [as in Eq. (10)] puts the plaquette in a coherent superposition thus renormalizing the rate of the majority move (by a factor $|c_\uparrow|^2$) and opening a new dissipative channel flipping the spin against the majority with a rate proportional to $\gamma_{\bar{\mu}}|c_\downarrow|^2$ according to the coefficients $c_\uparrow, c_\downarrow$ with $|c_\uparrow|^2 + |c_\downarrow|^2 = 1$.

dissipative lattice systems in more than one spatial dimension.

## 2 The model

We study the dissipative dynamics and steady-state of a quantum spin system on a square lattice. The dissipative process we consider implements a *majority vote* move on a plaquette

$$j\overset{\bullet}{\underset{\bullet\bullet}{}} = \{j, j + e_x, j + e_y\} \tag{1}$$

formed by the active spin $j$ at the vertex and by its Eastern ($j + e_x$) and Northern ($j + e_y$) neighbors, $e_{x,y}$ being two orthonormal versors. The spin at site $j$ aligns its $z$ component to the direction of the majority within the plaquette, if it is not already pointing in that direction, as pictorially shown in Fig. 1(a). This move can be represented by the action of the spin raising and lowering operators $\sigma_j^\pm = \frac{1}{2}(\sigma_j^x \pm \iota\sigma_j^y)$ ($\sigma_j^{x,y,z}$ being the Pauli matrices acting on the $j$-th site) conditioned on the majority within the plaquette. This process is encoded in the following jump operators

$$L_{j,\nu} = \sqrt{\gamma_\nu}\sigma_j^+ \mathbb{P}_{j\overset{\uparrow}{\underset{\bullet\bullet}{}}} \tag{2}$$

$$L_{j,\mu} = \sqrt{\gamma_\mu}\sigma_j^- \mathbb{P}_{j\overset{\downarrow}{\underset{\bullet\bullet}{}}}, \tag{3}$$

where $\mathbb{P}_{j}^{\uparrow/\downarrow}$ is the projector onto the respective set of states corresponding to up/down majority

$$\mathbb{P}_{j}^{\uparrow} = \frac{1}{4}\left(2 + \sum_{i \in j} \sigma_i^z - \prod_{i \in j} \sigma_i^z\right) \tag{4}$$

$$\mathbb{P}_{j}^{\downarrow} = \frac{1}{4}\left(2 - \sum_{i \in j} \sigma_i^z + \prod_{i \in j} \sigma_i^z\right). \tag{5}$$

The plaquette Eq.(1), then, defines the basic symmetry of the model; it is further *chiral*, as it lacks inversion symmetry.

Following Ref. [42], we consider noise on top of the exact majority moves implemented by Eqs. (2,3). These classical fluctuations correspond to *wrong* moves, where the spin aligns opposite to the majority in the plaquette

$$L_{j,\bar{\nu}} = \sqrt{\gamma_{\bar{\nu}}}\,\sigma_j^- \mathbb{P}_{j}^{\uparrow} \tag{6}$$

$$L_{j,\bar{\mu}} = \sqrt{\gamma_{\bar{\mu}}}\,\sigma_j^+ \mathbb{P}_{j}^{\downarrow}. \tag{7}$$

In the following, we fix the rate of the dissipative processes such that $\gamma_\mu + \gamma_{\bar{\mu}} = \gamma_\nu + \gamma_{\bar{\nu}} = \gamma$. It is further convenient to combine the decay rates introduced above into a dimensionless amplitude $T$ and bias $h$ defined as

$$T = \frac{\gamma_{\bar{\mu}} + \gamma_{\bar{\nu}}}{\gamma}, \quad h = \frac{\gamma_{\bar{\nu}} - \gamma_{\bar{\mu}}}{\gamma T}, \tag{8}$$

measuring the overall strength of classical fluctuations and quantifying the imbalance of the two noisy processes, respectively. When only dissipative processes are considered the dynamics can be exactly mapped onto the one of classical Ising-like models [51] where $T$ plays the role of an effective temperature and $h$ the one of a magnetic field.

In addition to the dissipative terms described above, we introduce quantum fluctuations through a Hamiltonian generating coherent evolution. The open-system dynamics of the system is then given by the Lindblad master equation [52] (hereafter we set $\hbar = 1$)

$$\dot{\rho} = -\imath[H, \rho] + \sum_{j,\alpha}\left(L_{j,\alpha}\rho L_{j,\alpha}^\dagger - \frac{1}{2}\{L_{j,\alpha}^\dagger L_{j,\alpha}; \rho\}\right), \tag{9}$$

with $\alpha = \nu, \mu, \bar{\nu}, \bar{\mu}$ labelling the different dissipative processes. As the plaquette chiral symmetry plays a central role in the purely dissipative case [42], we will investigate quantum fluctuations that break and preserve the chiral *plaquette symmetry*. In the following, we say that an operator has plaquette symmetry if it acts non-trivially over the full plaquette.

For this purpose, we investigate different paradigmatic Hamiltonians which represent different possible sources of quantum noise. The simplest Hamiltonian with no plaquette symmetry is given by a homogeneous transverse (with respect to dissipation) magnetic field

$$H_{\text{X}} = \Omega \sum_j \sigma_j^x, \tag{10}$$

which introduces independent single spin rotations.

As the dissipators inherently implement constraints through the majority within the plaquette, it is interesting to consider the interplay with kinetic constraints induced by the Hamiltonian. To this end, we introduce two different KCMs. (i) First, we study the two-dimensional

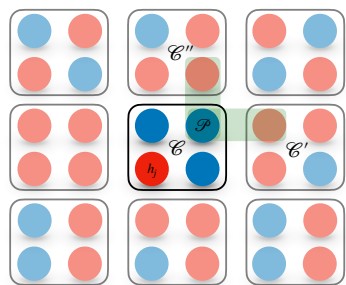

Figure 2: Sketch of the cluster mean-field approach. The spin lattice is partitioned into clusters (solid lines) that cover its entirety. The dynamics of a given cluster $\mathcal{C}$ are governed by the CMF Hamiltonian $H_{\mathrm{CMF}}$ and dissipator $\mathcal{L}_{\mathrm{CMF}}$ both including on-cluster and boundary terms. In this example we a show on-site Hamiltonian term $h_i$ and a plaquette interaction that couples $\mathcal{C}$ with $\mathcal{C}'$ and $\mathcal{C}''$.

version of the PXP model [53], implementing the Rydberg blockade on the square lattice with strength $\Omega_1$ and a Rydberg anti-blockade with strength $\Omega_2$

$$H^{\mathrm{PXP}} = \sum_j \Omega_1 \mathbb{P}^{\downarrow}_{j-e_x} \mathbb{P}^{\downarrow}_{j-e_y} \sigma^x_j \mathbb{P}^{\downarrow}_{j+e_x} \mathbb{P}^{\downarrow}_{j+e_y} + \Omega_2 \mathbb{P}^{\uparrow}_{j-e_x} \mathbb{P}^{\uparrow}_{j-e_y} \sigma^x_j \mathbb{P}^{\uparrow}_{j+e_x} \mathbb{P}^{\uparrow}_{j+e_y}. \tag{11}$$

Here $\mathbb{P}^{\uparrow\downarrow}_j = \frac{1}{2}(1 \pm \sigma^z_j)$ are the projectors onto the up and down spin state, respectively. (ii) Second, we introduce a NEC version of the PXP Hamiltonian defined above, where the projectors act only on the North and East neighboring sites

$$H^{\mathrm{PXP}}_{\bullet\bullet} = \sum_j \Omega_1 \mathbb{P}^{\downarrow}_{j+e_y} \sigma^x_j \mathbb{P}^{\downarrow}_{j+e_x} + \Omega_2 \mathbb{P}^{\uparrow}_{j+e_y} \sigma^x_j \mathbb{P}^{\uparrow}_{j+e_x}. \tag{12}$$

This Hamiltonian shares the chiral nature of the constraints implemented by the jump operators, thus introducing quantum fluctuations without breaking the plaquette symmetry of the NEC model [44].

# 3 Cluster mean-field approach

In this section we describe the CMF approach used in this work. For an extended introduction about this method the interested reader is referred to [46]. Here we briefly review the approach highlighting the peculiarities due to the the presence of multi-site Hamiltonian and dissipative terms acting on plaquettes of different shapes. We discuss both the inhomogeneous and translationally invariant version of CMF, used to compute time evolution of arbitrary initial states and the steady-state, respectively.

In full generality let us consider a Lindblad master equation $\dot{\rho} = -\imath[H, \rho] + \mathcal{L}[\rho]$ with Hamiltonian

$$H = \sum_j h_j + \sum_{\mathcal{P}} h_{\mathcal{P}}, \tag{13}$$

where $h_j$ encodes single-site spin terms and $h_{\mathcal{P}}$ takes into account multi-spin interactions of contiguous sites belonging to the plaquette $\mathcal{P}$. For instance, if we focus on the Hamiltonian (12) we have that

$$\mathcal{P} = \left\{ {}_j \bullet_{\bullet}^{\bullet} \,|\, j \in \text{lattice sites} \right\} \tag{14}$$

is the set of NEC plaquettes (1). Let us also consider the following dissipators

$$\mathcal{L}[\rho] = \sum_j \mathcal{L}_j[\rho] + \sum_{\mathcal{P}'} \mathcal{L}_{\mathcal{P}'}[\rho], \tag{15}$$

where $\mathcal{L}_j[\rho] = l_j \rho l_j^\dagger - \{l_j^\dagger l_j; \rho\}/2$ and $\mathcal{L}_{\mathcal{P}'}[\rho] = l_{\mathcal{P}'} \rho l_{\mathcal{P}'}^\dagger - \{l_{\mathcal{P}'}^\dagger l_{\mathcal{P}'}; \rho\}/2$ account for incoherent processes acting on the $j$-th site and on the plaquette $\mathcal{P}'$. Again $l_{\mathcal{P}'}$ is a jump operator acting on contiguous sites forming the plaquette $\mathcal{P}'$. For the dissipative NEC processes considered in this work we have that $\mathcal{P}' = \mathcal{P}$ defined in Eq. (14).

Let us now consider the CMF ansatz for the system density matrix

$$\rho_{\mathrm{CMF}} = \bigotimes_{\mathcal{C}} \rho_{\mathcal{C}}, \tag{16}$$

where $\rho_{\mathcal{C}}$ is the density matrix of the $\mathcal{C}$-th cluster. This framework allows to systematically go beyond the single-site Gutzwiller approximation considering spatially extended clusters composed by many sites. In this work we will consider square-shaped clusters composed by $\ell \times \ell$ spins. Within this approach short-range correlations are taken into account exactly within the cluster $\mathcal{C}$ while those among neighboring clusters are treated at a mean-field level. Inserting the ansatz (16) into the master equation we get

$$\dot{\rho}_{\mathcal{C}} = -\iota[H_{\mathrm{CMF}}, \rho_{\mathcal{C}}] + \mathcal{L}_{\mathrm{CMF}}[\rho_{\mathcal{C}}], \tag{17}$$

where $H_{\mathrm{CMF}} = H_{\mathcal{C}} + H_{\mathcal{B}(\mathcal{C})}$ and $\mathcal{L}_{\mathrm{CMF}} = \mathcal{L}_{\mathcal{C}} + \mathcal{L}_{\mathcal{B}(\mathcal{C})}$ are the CMF Hamiltonian and dissipator, respectively. The on-cluster part includes Hamiltonian terms whose support lies entirely within the cluster $\mathcal{C}$

$$H_{\mathcal{C}} = \sum_{j \in \mathcal{C}} h_j + \sum_{\mathcal{P} \in \mathcal{C}} h_{\mathcal{P}}. \tag{18}$$

The boundary term can be written as

$$H_{\mathcal{B}(\mathcal{C})}(t) = \sum_{\mathcal{P} \in \mathcal{B}(\mathcal{C})} \mathrm{Tr}_{\tilde{\mathcal{C}} \neq \mathcal{C}}[\rho_{\mathrm{CMF}}(t) h_{\mathcal{P}}], \tag{19}$$

where $\mathcal{B}(\mathcal{C})$ is the boundary of the cluster and represents the mean-field interactions among different clusters induced by Hamiltonian plaquette terms $\mathcal{P}$ whose support lies partially outside the cluster $\mathcal{C}$. The trace in Eq. (19) is thus taken over the neighboring clusters $\tilde{\mathcal{C}}$ and gives an operator with support entirely on $\mathcal{C}$ multiplied by a time dependent field that needs to be computed self-consistently in time. The structure of the dissipative part mirrors the unitary one where the on-cluster dissipator $\mathcal{L}_{\mathcal{C}}$ includes local and plaquette jump operators with support entirely on the cluster and the boundary terms

$$\mathcal{L}_{\mathcal{B}(\mathcal{C})}(t)[\rho_{\mathcal{C}}] = \sum_{\mathcal{P}' \in \mathcal{B}(\mathcal{C})} \mathrm{Tr}_{\tilde{\mathcal{C}}' \neq \mathcal{C}}[\mathcal{L}_{\mathcal{P}'}[\rho_{\mathrm{CMF}}(t)]] \tag{20}$$

take into account dissipative process on the plaquette $\mathcal{P}'$ acting on the boundary $\mathcal{B}(\mathcal{C})$ whose support is shared between $\mathcal{C}$ and the surrounding clusters. In this case the trace provides a new set of jump operators with support on $\mathcal{C}$ multiplied by a time-dependent rate.

As a result the full dynamics are eventually simplified into the coupled reduced dynamics of the clusters (see Fig. 2) that can be computed numerically or analytically (for small-size clusters). In terms of complexity the CMF approach requires to compute the evolution of $\mathcal{O}(Md^{\ell^2})$ coupled nonlinear differential equations where $M$ is the number of clusters that cover the lattice (made of $N = M\ell^2$ sites), $d$ is the local Hilbert space dimension ($d = 2$ for spin-1/2) and $\ell^2$ is the number of sites composing each cluster. This is the case if all the cluster density matrices are distinct, a condition needed when the dynamics of spatially inhomogeneous states are considered. However, this condition can be relaxed when we consider translationally invariant states imposing $\rho_{\mathcal{C}} = \rho_{\mathcal{C}'}, \forall \mathcal{C}, \mathcal{C}'$. This leads to a simplified version of the CMF ansatz where only the evolution of a single representative cluster is needed [resulting into $\mathcal{O}(d^{\ell^2})$ equations]. In this case the thermodynamic limit $N \to \infty$ is implicitly taken and the steady-state is defined as

$$\rho_{ss} = \lim_{t \to \infty} \rho_{\mathcal{C}}(t). \tag{21}$$

# 4 Steady-state phase diagram

We study the phase diagram of the NEC model in the parameter space defined by the amplitude of classical fluctuations $T$, their bias $h$ and by the amplitude of quantum fluctuations $\Omega$. In the rest of the paper the Hamiltonian couplings will be expressed in units of $\gamma$. In particular, to establish the presence of bistability we perform hysteresis cycles on the bias $h$, adiabatically sweeping $h$ forwards and backwards from $-1 \rightarrow 1$ using a small increment $dh = 0.1$. For each point $(\Omega, T)$, we evolve according to Eq. (17) an initial state corresponding to the steady-state at the previous value of $h$ $\rho_0(h) = \rho_{ss}(h \pm dh)$, until the steady-state is reached $\dot{\rho}_{ss}(h) = 0$.

To probe bistability, we focus on the magnetization in the steady-state, $m_z = \text{Tr}[\rho_{ss} \sum_j \sigma_j^z / \ell^2]$. Within the bistable region, the forwards and backwards sweeps over $h$ result in different steady-state magnetization $m_z^{(f,b)}$. We then define the order parameter $\Delta m_z = |m_z^{(f)} - m_z^{(b)}|$, which captures the presence of bistability.

As we show below, the results concerning the boundaries of the bistable region are already converged when comparing the CMF results for clusters of size $\ell \times \ell$ with $\ell = 2$ and $\ell = 3$, thus suggesting that the results at $\ell = 3$ capture all relevant correlations.

## 4.1 Robustness to different Hamiltonians

To understand the effect of quantum fluctuations on bistability, we compare three fundamentally different cases. First, we use the NEC-symmetric PXP Hamiltonian Eq. (12), which shares the same plaquette symmetry of the dissipators. As such, this Hamiltonian is expected to have weaker effects on the bistable phase. We then break the plaquette symmetry via the 2d PXP Hamiltonian (11), which however retains the constrained structure. Finally, we study the effect of completely relaxing the constraint on neighboring sites, applying Eq. (10), which performs free rotations about the $x$-axis.

As mentioned above, for each of these models we perform hysteresis cycles and compare the magnetization in the forwards and backwards sweep. In Fig. 3(a) we present the steady-state magnetization $m_z$ for the NEC-symmetric Hamiltonian at fixed $\Omega_1 = \Omega_2 = \Omega = 0.1$ and for different amplitudes of the classical noise $T \in [0.1, 0.25]$ [1]. In this range of parameters, bistability is revealed by the different values of $m_z^{(f)}$ and $m_z^{(b)}$, indicating the presence of two steady-states. As the amplitude $T$ is increased (blue to red), the bistable region progressively shrinks, until eventually it disappears for $T \gtrsim 0.25$.

Using the difference in steady-state magnetization $\Delta m_z$ as an order parameter, we can further obtain a phase diagram distinguishing the normal phase $\Delta m_z = 0$ from the bistable phase $\Delta m_z > 0$. We show such a phase diagram for the NEC Hamiltonian in Fig. 3(b). At fixed $\Omega$ the bistable phase is symmetric around $h = 0$, and shrinks as $T$ is increased. The bistable phase shrinking follows a phase boundary defined by the phenomenological $T$-dependent critical bias

$$h_c(T) = \pm \left( 1 - \frac{\sqrt{T - T^*}}{R} \right), \tag{22}$$

as shown by the red dashed line fitting the data for $h_c$. Here $T^* = T_c(|h| = 1)$ and $T_c(h = 0)$ define the critical temperatures at $|h| = 1$ and $h = 0$, respectively, and specify the value of $R = \sqrt{T_c(h = 0) - T^*}$. We also note that $T^*$ corresponds to the largest $T$ such that the system is bistable for all values of $h$. These two effective temperatures depend on the strength of the Hamiltonian coupling, for $\Omega = 0.1$ as in Fig. 3 we get $T^* \approx 0.08$ and $T_c(h = 0) \approx 0.25$.

Taking a closer look at the decay of the order parameter $\Delta m_z$ reveals an interesting behavior distinguishing the transition at $h = 0$ from the rest. As we show in Fig. 4(a), at all non-zero values of $h$ the order parameter seems to present a discontinuous transition at $T = T_c(h)$, with

---

[1]The results are qualitatively similar for all other models studied.

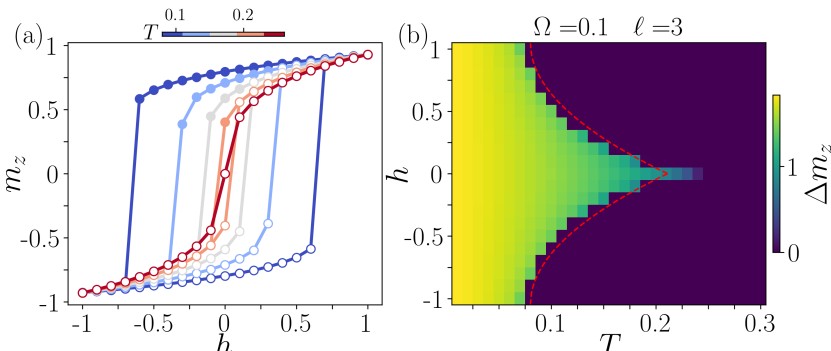

Figure 3: (a): Hysteresis curves for the steady-state magnetization of the constrained plaquette model Eq. (12) show clear signatures of bistability. The forwards (full circles) and backwards (empty circles) curves are different over a large region of bias at various amplitudes $T$, defining an area of bistability. (b): Using $\Delta m_z$ as an order parameter, we obtain the phase diagram of the NEC model at fixed $\Omega$. The boundaries of the bistable phase are captured by the formula (22) (red dashed line).

a finite $\Delta m_z$ within the bistable phase suddenly vanishing as the critical point is crossed. On the other hand, at $h = 0$ this is not the case, and the order parameter gradually decreases, $\lim_{T \to T_c^-} \Delta m_z = 0$. Our finite $\ell$ analysis then suggests that the NEC model hosts two different phase transitions, a first order transition at $h \neq 0$ and a continuous phase transition in absence of bias. A conclusive distinction of the two transitions, however, would require the investigation of much larger cluster sizes, beyond current computational availability. Nonetheless, a continuous transition at $h = 0$ was already pointed out in the classical case [42].

We now explore the effect of increasing the quantum fluctuation amplitude $\Omega$ by analyzing the curves $T_c(h)$, as shown in Fig. 4(b). In analogy with the phase diagram shown in Fig. 3(b), the critical amplitude curves are symmetric around $h = 0$ and show a robust quadratic behavior

$$T_c(h) = R^2 (1 - |h|)^2 + T^*, \tag{23}$$

obtained inverting Eq. (22). As we increase $\Omega$, the curves shift to lower values, indicating a shrinking of the bistable phase as quantum fluctuations become stronger. This is in agreement with the expectation that coherent dynamics will eventually destroy bistability. However, the bistable region persists up to considerable values of $\Omega \approx 0.25$.

We now check the convergence of the CMF ansatz by comparing the behavior of the system at increasing cluster sizes $\ell$. In Fig. 5 we show that the critical curves for $\Omega$ [Fig. 5(a)] and $T$ [Fig. 5(b)] do not change as the cluster size is increased. We compare the results for $\ell = 2, 3$ obtained evolving the density matrix with results for $\ell = 4$ averaged over $N = 500$ trajectories, with a standard deviation corresponding to the error bars in the figure. The comparison between $\ell = 2$ (dashed lines), $\ell = 3$ (solid lines) and $\ell = 4$ (dotted lines) shows good convergence of our numerical simulations, thus suggesting that the bistable phase we observe is stable with respect to the CMF approximation [46, 47].

Qualitatively similar results hold for all other Hamiltonians defined in Sec. 2, as shown in the Appendix. This highlights the universality of bistability in the NEC model and its stability towards different microscopic coherent processes. However, the details of the model inducing quantum fluctuations impact dramatically on the extension of the bistable phase. As we show in Fig. 5, the model shapes the critical curves, both for $\Omega$ (c) and $T$ (d).

We can understand these differences considering that the Hamiltonians (10), (11) and (12) all induce local rotations about the $x$-axis of the Bloch sphere, thus inducing an effective depolarization of the $z$ component of the spin. This process progressively destroys bistability and

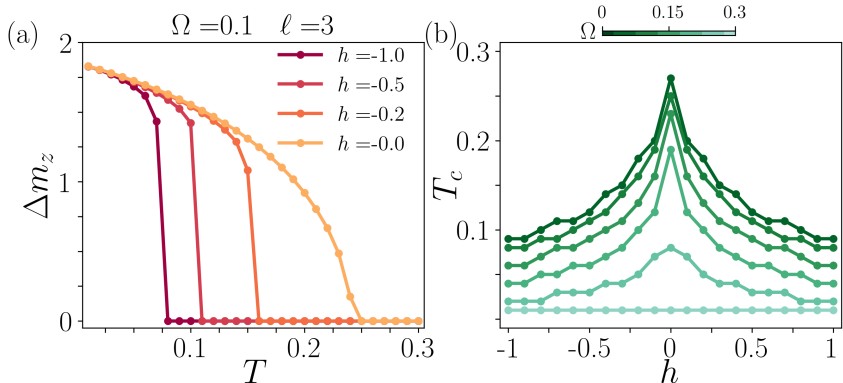

Figure 4: (a): Analyzing the decay of $\Delta m_z$ at different values of the bias highlights that at $h = 0$ $\Delta m_z \to 0$ as the amplitude reaches its critical value $T_c$, thus suggesting a continuous phase transition. On the other hand, at $h \neq 0$ $\Delta m_z$ attains a finite value at the critical point, indicating a first order transition. (b): The critical amplitude $T_c$ as a function of $h$ for different values of $\Omega$. Similarly to the phase diagram, the critical lines are symmetric around $h = 0$ and show a quadratic behavior as in Eq. (23). As the strength of quantum fluctuations is increased (dark green to light blue), the critical amplitude monotonically decreases.

plays a role similar to an effective temperature. However, in the three classes of Hamiltonians considered the action of $\sigma^x$ is constrained by the status of a certain number of neighbors. Precisely, $H_X$ is unconstrained, while $H^{\mathrm{PXP}}$ and $H^{\mathrm{PXP}}$ are constrained by two and four nearest neighbors, respectively. As a consequence, the depolarizing action is stronger as the number of constrains is smaller. This is in agreement with the numerical results in Fig. 5 (c),(d).

In conclusions, these results teach us that the microscopic details of the Hamiltonian are not a fundamental ingredient to determine the qualitative structure of the steady-state phase diagram, yet they are crucial in determining the extension of the phases. Following this line of reasoning in the next section we test the robustness with respect to additional dissipative processes.

## 4.2 Stability of the bistable region under unconstrained dissipation

Driven by the results of the previous section we now want to understand if the structure of the steady-state phase diagram is stable against the presence of dissipative processes beyond the NEC ones.

We choose the simplest jump operator with no kinetic constraints and no plaquette symmetry

$$L_{j,x} = \sqrt{\Gamma}\sigma_j^x. \tag{24}$$

We then study the steady-state resulting from the purely dissipative dynamics generated by the NEC operators Eqs. (2-3,6-7) together with the dissipative free rotations along $x$ given by $L_{j,x}$.

The dissipative perturbation affects bistability in a qualitatively similar way than the quantum fluctuations presented in the previous Section. As we show in the Appendix, the bistable phase shares the same features as in the Hamiltonian case. However, comparing quantitative results from the action of $L_{j,x}$ with the one of $H_X$ shows a dramatic effect of replacing coherent with dissipative spin rotations. As we show in Fig. 5 (c),(d), the dissipative perturbation has a much weaker effect on bistability than its Hamiltonian counterpart.

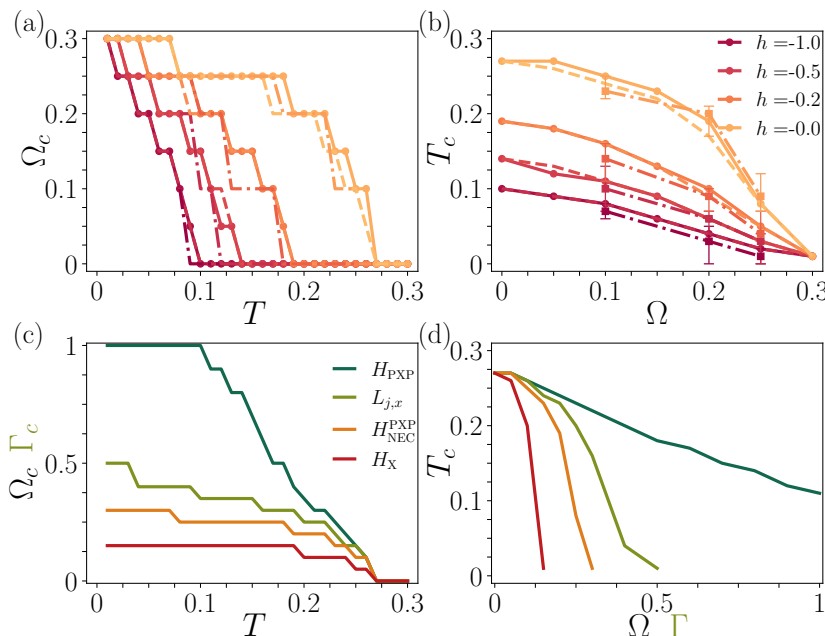

Figure 5: (a): Critical amplitude of quantum fluctuations $\Omega_c$ as a function of $T$ for different biases. Comparison of $\ell = 2$ (dashed lines) with $\ell = 3$ (solid lines) and $\ell = 4$ (dotted lines) suggests convergence of the cluster mean-field approach. (b): Critical amplitude of classical noise $T_c$ as function of $\Omega$. Again comparison of $\ell = 2, 3$ and $4$ suggests convergence of CMF. Comparison of the curves for $\Omega_c$ (c) and $T_c$ (d) at $h = 0$ for different representative models. The bistable phase is strongest in the case of the PXP Hamiltonian. This suggests that kinetic constraints play an important role, as they weaken the dephasing action of $\sigma^x$. We further compare dissipative and coherent free independent rotations of each spin. The dissipative action of $\sigma^x_j$ has a weak effect on bistability, while its coherent counterpart results in a quick destabilization of the bistable phase.

This result highlights the robustness of the phase diagram with respect to additional dissipation sources. The fact that we are not in a fine-tuned situation is important also for possible concrete implementation (for example with Rydberg atoms) where the presence of incoherent on-site dephasing is unavoidable.

# 5   Stability Analysis

To gain more insight into the phase diagram presented in the previous section, and to provide further proof of convergence of our CMF simulations, we perform a stability analysis [46, 54]. This procedure consists in considering fluctuations on top of the cluster mean-field steady-state and investigating their dynamics. Whenever fluctuations grow uncontrolled, the system is unstable, whereas if fluctuations are reabsorbed, the steady-state is converged.

To this aim, let us consider the state of the $n$-th cluster (located at position $\mathbf{r}_n$)

$$\rho^{(n)} = \rho_{ss} + \delta\rho^{(n)} = \rho_{ss} + \sum_{\mathbf{k}} e^{i\mathbf{k}\cdot\mathbf{r}_n}\delta\rho_{\mathbf{k}}, \tag{25}$$

where $\rho_{ss}$ is the homogeneous steady-state reached within the CMF approach (21) and $\delta\rho^{(n)}$ are small spatial fluctuations expanded in plane waves with wavevector $\mathbf{k} = (k_x, k_y)$. The

wavevectors are limited to the region $[-\pi/\ell, \pi/\ell]$, as they are restricted to the first Brillouin zone of the superlattice with unit cell given by the $\ell \times \ell$ cluster.

In the cluster mean-field approach, the Liouvillian super operator is split into on-cluster and boundary terms. The latter are written as the product of operators with support on the $n$-th cluster, and others with support on the neighboring one in direction $q$: $H_{\mathcal{B}} = \sum_{j\in\mathcal{B}} h_j^{(n)} h_{j+e_q}$, $\mathcal{L}_{\mathcal{B}} = \sum_{j\in\mathcal{B}} l_j^{(n)} l_{j+e_q} \cdot l_{j+e_q}^{\dagger} l_j^{(n)\dagger} - \frac{1}{2}\{l_{j+e_q}^{\dagger} l_j^{(n)\dagger} l_j^{(n)} l_{j+e_q}; \cdot\}$. These operators are then evaluated on the corresponding cluster wavefunction, and therefore the boundary terms are non-linear super operators on the state $\rho$.

We now write the equation of motion for the state in Eq. (25)

$$
\dot{\rho}^{(n)} = \left(-\imath[H_{\mathcal{C}}, \cdot] + \mathcal{L}_{\mathcal{C}}\right)\left[\rho_{ss} + \delta\rho^{(n)}\right] + \sum_{\substack{q=x,y \\ j\in\mathcal{B}_q}}\Bigg[ -\imath\,\mathrm{tr}\left[h_{j+e_q}(\rho_{ss} + \delta\rho^{(n)})\right]\left[h_j, \rho_{ss} + \delta\rho^{(n)}\right]
$$
$$
+ \left|\mathrm{tr}\left[l_{j+e_q}(\rho_{ss} + \delta\rho^{(n)})\right]\right|^2 \mathcal{L}_j^{(n)}\left[\rho_{ss} + \delta\rho^{(n)}\right]\Bigg] + \text{corner term},
$$

(26)

where the corner term refers to the top right site in the cluster, where both the top and the right neighboring clusters contribute to the expectation values; its expression, together with details on the derivation of the equations for the stability analysis, are reported in Appendix C. We group the on-cluster terms with all the terms in the sum where the expectation values are evaluated on the steady-state, which results in the cluster mean-field Liouvillian acting trivially on the steady-state itself, $\mathcal{M}_{\mathrm{CMF}} = -\imath[H_{\mathrm{CMF}}, \cdot] + \mathcal{L}_{\mathrm{CMF}}$, $\mathcal{M}_{\mathrm{CMF}}[\rho_{ss}] = 0$. We further linearize the remaining part of the equation of motion, neglecting all non-linear terms in $\delta\rho^{(n)}$. Using the plane waves expression of the fluctuations we get $\mathrm{tr}[O_{j+e_q}\sum_{\mathbf{k}} e^{\imath\mathbf{k}\cdot\mathbf{r}_{n+e_q}}\delta\rho_{\mathbf{k}}] = \sum_{\mathbf{k}} e^{\imath k_q}\mathrm{tr}[O_{j+e_q}\delta\rho_{\mathbf{k}}]$. As we are working under the translationally invariant assumption, the expectation value above corresponds to the respective operator evaluated on cluster $n$, and we drop the superscript $n$ hereafter. The equation of motion then becomes,

$$
\dot{\delta\rho}_{\mathbf{k}} = \mathcal{M}_{\mathrm{CMF}}[\delta\rho_{\mathbf{k}}] + \sum_{\substack{q=x,y \\ j\in\mathcal{B}_q}} e^{\imath k_q}\Bigg[ -\imath\left[h_j, \rho_{ss}\right]\mathrm{tr}\left[h_{j+e_q}\delta\rho_{\mathbf{k}}\right]
$$
$$
+ 2\Re\left[\mathrm{tr}\left[l_{j+e_q}\rho_{ss}\right]\right]\mathcal{L}_j^{(n)}[\rho_{ss}]\mathrm{tr}\left[l_{j+e_q}\delta\rho_{\mathbf{k}}\right]\Bigg] + \text{corner term}.
$$

(27)

Finally, we notice that the expectation values can also be thought of as the action of a super operator on the state $\delta\rho_{\mathbf{k}}$, and, upon vectorization, they correspond to a vector in the enlarged Hilbert space $\langle\!\langle h_{j+e_q}\|$, $\langle\!\langle l_{j+e_q}\|$. Similarly, the commutator and dissipator with the steady-state also become vectors, $\|\rho_{h_j}\rangle\!\rangle$, $\|\rho_{\mathcal{L}_j}\rangle\!\rangle$, and the sum in the equation above corresponds to a sum of rank-1 matrices given by the outer product of these vectors

$$
\partial_t\|\delta\rho_{\mathbf{k}}\rangle\!\rangle = \Bigg[\mathbb{M}_{\mathrm{CMF}} + \sum_{\substack{q=x,y \\ j\in\mathcal{B}_q}} e^{\imath k_q}\bigg(-\imath\|\rho_{h_j}\rangle\!\rangle\langle\!\langle h_{j+e_q}\| + 2\Re\left[\mathrm{tr}\left[l_{j+e_q}\rho_{ss}\right]\right]\|\rho_{\mathcal{L}_j}\rangle\!\rangle\langle\!\langle l_{j+e_q}\|\bigg)
$$
$$
+ \text{corner term}\Bigg]\|\delta\rho_{\mathbf{k}}\rangle\!\rangle.
$$

(28)

In order to evaluate the behavior of fluctuations at a given value of $\mathbf{k}$, we diagonalize the matrix governing their dynamics according to Eq. (28) for a cluster of $2 \times 2$ sites. In particular, the eigenvalue with the largest real part $\mu_k$ determines whether fluctuations will grow, stay

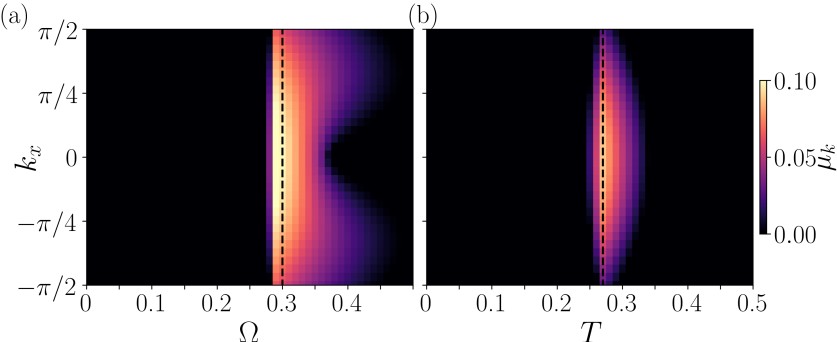

Figure 6: The largest eigenvalue of the super operator $\mathcal{M}_\mathbf{k}$ at fixed $k_y = 0$ for a $2 \times 2$ cluster. (a): Driving the transition with quantum fluctuations at $T = 0$ (hence $h = 0$) yields a very sharp spreading of instability close to the critical point (vertical dashed line), spawning from $k_x = 0$ to all wavevectors and peaking at $|k_x| = \pi/4$. (b): Classical fluctuations at $\Omega = 0$ and $h = 0$ present a similar behavior in the vicinity of the critical point, although narrower and with no spatial structure, as the instable region is peaked at $k_x = 0$.

stable, or vanish in time. Since for an $\ell \times \ell$ cluster the components of the vectors $\mathbf{r}_n$ must be $\ell$ times the elementary lattice vectors, the range of lattice momenta allowed are restricted to the first Brillouin zone of the superlattice, $|k_{x,y}| < \pi/\ell$. Physically this means that we are considering fluctuations with at most the periodicity of the cluster structure since shorter ones $|k_{x,y}| > \pi/\ell$ are already included exactly within the CMF ansatz.

As we show in Fig. 6 (where $k_y = 0$), $\mu_k = 0$ within the bistable phase, while it becomes positive around the critical point, both when the transition is driven by quantum fluctuations $\Omega$ (a) and by the amplitude $T$ (b).

The presence of $\mu_k > 0$ in the vicinity of the transition (dashed lines) is related to criticality. Indeed as we approach the critical point, fluctuations can grow and drive the system from one of the two steady-states of the bistable phase in the unique one of the normal phase. Since the steady-state is translationally invariant in both the phases, this process triggers an instability at $\mathbf{k} = 0$. A positive $\mu_k$ at criticality suggests that larger cluster sizes are needed to precisely pinpoint the critical point, due to the build-up of long range correlations.

Interestingly, the critical region where $\mu_k > 0$ features a different structure depending on whether the transition is driven by quantum or thermal effects. In the former case [panel(a)] the instability region is wider close to the criticality and shows two pronounced peaks at $|k_x| = \pi/4$. In the latter situation [panel(b)] the instability region is narrower, featuring a persistent instability at $k_x = 0$ while larger wavevectors get progressively stable as $T$ is increased.

Away from critical points the cluster mean field solution is stable at all wave vectors, indicating its convergence. In the bistable phase, this corresponds to $\mu_k = 0$ due to the presence of 2 distinct steady states. In fact, one can write $\rho^{(n)} = \rho_{ss}^{(1)} + \rho_{ss}^{(2)}$, where $\rho_{ss}^{(1)}$ and $\rho_{ss}^{(2)}$ are the two steady states. As such, the state $\rho^{(n)}$ will be stationary itself. It then follows that the super-operator describing the dynamics of fluctuations on top of one steady state admits zero eigenvalues.

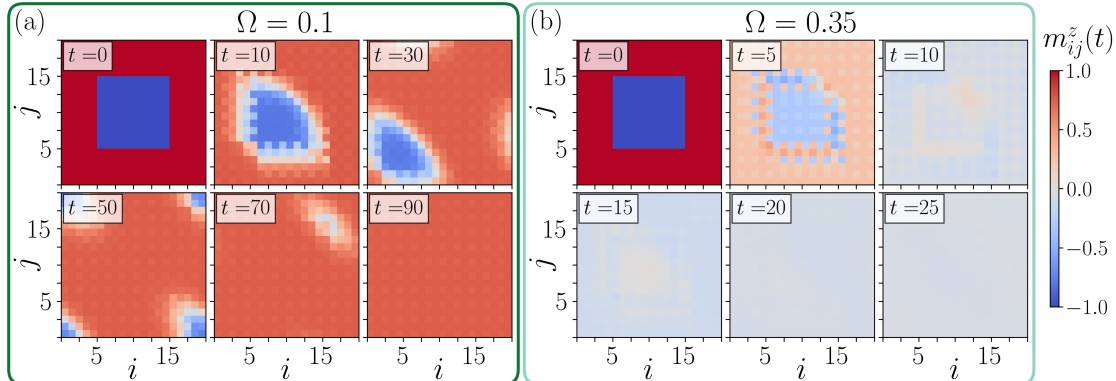

Figure 7: Dynamics of a square island of $\ell_\downarrow \times \ell_\downarrow$ down spins in a square lattice of $2\ell_\downarrow \times 2\ell_\downarrow$ spins, with $\ell_\downarrow = 10$. We fix the amplitude to $T = 0.1$ and the bias to $h = 0.1$, and tune the bistability by the value of $\Omega$. (a): Within the bistable region the bubble is always absorbed by the surrounding phase, this mechanism allows the presence of two distinct steady-states. From this initial configuration, the steady-state magnetization is large and positive, in spite of $h > 0$ favoring negative magnetization. (b): When bistability is lost the bubble mixes with its surroundings, spreading to the entire lattice and destroying the initial order. In the normal phase the global magnetization is small and negative in the steady-state.

## 6 Nonergodic dynamics

Finally, we use the iCMF method introduced above to investigate the dynamics of large square lattices. We focus on initial states where a minority island, i.e. a square region composed of $\ell_\downarrow \times \ell_\downarrow$ spins aligned with the bias, is embedded in a sea of spins pointing in the opposite direction.

In the normal phase, there is a unique steady-state, and the island is expected to fade away as the system evolves over long timescales. In this case, no information about the initial conditions is retained at long times. In the bistable phase, instead, the fate of the island is a priori unclear since the two steady-states are very close to the possible alignments. In this case the role of the interface is crucial and the surface tension determines the dynamics of the island.

The key physical mechanism enabling the bistability of the NEC model is the presence of chiral dissipative processes. If the symmetry of the jump operators is restored by conditioning the flip of the central spin to the majority of all the four nearest neighbor sites, the bistability region disappears and the disspative only dinamics can be mapped onto a classical Ising system below its critical point [51]. Here a circular bubble of radius $r$ composed by spins of one of the two phases evolves as [55]

$$\boxed{\text{Class. Ising}} \quad \frac{\mathrm{d}r}{\mathrm{d}t} = -\frac{A}{r} + B|h| \tag{29}$$

where the first term induces the absorption ($A > 0$) of the bubble with a negative speed proportional to $1/r$ and the second term is induced by a a small symmetry-breaking bias $|h| \ll 1$ that favors ($B > 0$) or disfavors ($B < 0$) energetically the phase inside the bubble. If $B > 0$ Eq. (29) admits a critical radius $r_c \propto 1/h$, for any arbitrary small bias $h$, above which the bubble will expand indefinitely and will take over the entire system. At a classical level this scenario drastically changes for the NEC model [42] due to the angular dependence of the

surface tension caused by the peculiar plaquette structure. In this case we get [56]

$$\boxed{\text{Class. NEC}} \quad \frac{\mathrm{d}r}{\mathrm{d}t} = -C + B|h| \tag{30}$$

with $C > 0$. Here an arbitrary small bias is not enough to allow for the expansion and the bubble is reabsorbed at approximately constant velocity, regardless of its radius. The classical statistical mechanics of the interfaces shows that the Ising model displays metastability and features an ergodic behaviour while the classical NEC model breaks ergodicity.

In order to understand how the NEC behavior is modified in the presence of quantum fluctuations, we simulate the dynamics of a square lattice of linear dimension $L = 20$, obtained through iCMF with $\ell = 2$. We fix the amplitude and bias to $T = h = 0.1$ and we tune between the normal and the bistable phase using the amplitude of quantum fluctuations, $\Omega$, induced by the Hamiltonian (12). As $h > 0$ the island corresponds to down-spins [$B > 0$ in Eq. (30)]. We analyze the dynamics of islands of various size $\ell_\downarrow$.

In Fig. 7, we show the dynamics of a large island, $\ell_\downarrow = 10$, both in the bistable [panel (a)] and in the normal phase [panel (b)]. In the bistable phase, the island gets reabsorbed at a certain velocity $v$ by the surrounding down-spins. Due to the shape of the island this process follows a precise pattern, starting from the top right corner and following the diagonal to the bottom left corner. In the normal phase, $\Omega = 0.35$, instead, the island disappears in favor of the unique steady. It is worth noticing the faster equilibration timescale of the normal phase, as compared to the bistable phase. This is expected since the relaxation timescale in bistable region is determined by the reabsorption physics that takes place at the boundary of the island and thus depends on its size.

A more thorough analysis reveals a dramatically different scaling for the relaxation timescale, $\tau$, in the two phases as the island size is changed. As we show in Fig. 8 (a), $\tau$ increases linearly with the bubble size $\ell_\downarrow$ in the bistable phase. This implies that the reabsorption velocity is independent from the island size [as in the classical case, see Eq. (30)] and can be extracted by fitting the data with $\tau = \sqrt{2}\ell_\uparrow/v$. Interestingly, by performing a scaling analysis of the reabsorbtion velocity [Fig. 8 (b)] we find that both the quantum fluctuations $\Omega$ and the effective temperature $T$ reduce the velocity. For small $\Omega$ and $T$ we find that such reduction is linear and the contributions from the the two sources of fluctuations are independent. Thus Eq. (30) becomes

$$\boxed{\text{QNEC}} \quad \frac{\mathrm{d}r}{\mathrm{d}t} = -C + Bh + DT + E\Omega \tag{31}$$

with $D, E > 0$ that quantify the impact of thermal and quantum fluctuations at the lowest order, respectively. From our analysis we find that

$$D \approx 2.4 \pm 0.2 \quad \text{and} \quad E \approx 0.25 \pm 0.04, \tag{32}$$

hinting for a weaker effect of quantum fluctuations with respect to the thermal ones. Notice that the intercept of the two curves in the inset of Fig. 8(b) is different, as for $T$ the intercept is given by $-C + Bh + E(\Omega = 0.1)$, while for $\Omega$ by $-C + Bh + D(T = 0.1)$. The difference in the intercept is then given by $0.1(D - E) \approx 0.2$, compatible with the one observed numerically and consistent with Eq. (31).

Approaching the critical point the absorption capacity of the system is weakened and eventually the whole lattice tends to align with the direction favored by $h$ (negative magnetization). As a consequence, for small $\ell_\downarrow$ the system is very far from the steady-state, and $\tau$ gains an inverse proportionality to the size of the island as shown in Figure 8 (a) for $\Omega = 0.2$. Deep in the normal phase $\tau$ becomes approximately constant with respect to $\ell_\downarrow$. This suggests a bulk mechanism responsible for the dissolution of the island that does not involve the physics of the boundary.

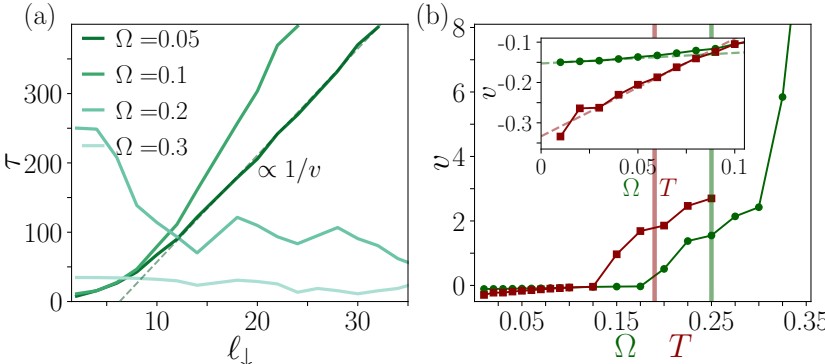

Figure 8: (a): Timescale $\tau$ for the full relaxation of the island at $T = h = 0.1$. In the bistable phase $\tau$ grows proportionally to the linear size of the island, $\ell_\downarrow$ and increases as $\Omega$ becomes larger. As $\Omega$ approaches the critical point, the linear increase with $\ell_\downarrow$ is lost and eventually, in the normal phase $\tau$ becomes constant. In the bistable phase, we determine the island velocity from the slope of the linear fit $\tau \approx \sqrt{2}\ell_\uparrow/v$ (dashed line). (b): The reabsorption velocity is a monotonically increasing function of $\Omega$ and $T$. Data are obtained at fixed $T = h = 0.1$ (green curve) and $\Omega = h = 0.1$ (red curve). The shaded areas represent the critical value of $\Omega$ and $T$. Close to the transition, the velocity becomes positive, as the system is not able to absorb the island anymore. In the inset, we show a zoom-in of the small $\Omega, T$ region. In this regime, the velocity increases linearly as a function of $\Omega$ and $T$ as predicted in Eq. (31), and we extract the constants $D$ and $E$ from a linear fit (dashed lines).

# 7 Conclusions and perspectives

In this work we explored the physics of the dissipative quantum North-East-Center model for spin-1/2 on a square lattice. Exploiting extensive numerical calculations based on the cluster mean-field approach we computed the steady-state phase diagram of the model in the presence of different classes of Hamiltonians and competing dissipators.

The phase diagram hosts a bistable phase, featuring two steady-states with opposite macroscopic magnetization, and a normal phase with a unique steady-state. The transition between these two phases is driven by thermal and quantum fluctutations, which eventually destroy the mechanism giving rise to bistability. We defined a suitable order parameter for the bistable phase, and we located the critical lines of the phase diagram. The convergence of the phase boundaries with respect to the size of the clusters used in the ansatz ensures the accuracy of our results.

Our findings indicate that the structure of the phase diagram is universal with respect to the Hamiltonian dynamics and robust when an additional single-site dephasing channel is added. This suggests that chiral kinetically constrained dissipative models could provide a robust mechanism for genuine bistability in quantum many-body systems, beyond long-range systems [57, 58]. The linear stability analysis corroborates the validity of the cluster ansatz and shows that the instabilities triggered at the critical points feature non-trivial directional dependence, able to distinguish if the transition is induced by thermal or quantum fluctuations. Using the inhomogeneous version of the cluster mean-field ansatz, we also investigated the dynamical emergence of bistability. In analogy with the bubble absorption and proliferation in Ising-like models, we analyzed the dynamics of initial states where islands of spin point in the opposite direction with respect to the background. We found that bistability is rooted in the capacity of the system of absorbing islands of arbitrary sizes, which persists in presence of

coherent Hamiltonian dynamics, confirming robust bistability also in the quantum setting. We characterized such dynamics and proposed an equation of motion for the islands reabsorption velocity that captures the effect of coherent quantum dynamics at linear order. We further showed that in the normal (non-bistable) phase the boundary mechanism leading to island reabsorption is completely absent, and thus a bulk mechanisms is responsible for the relaxation to the (unique) steady-state.

This particular feature of the NEC model can provide an interesting route to stabilize quantum states against random noise resulting in rare regions of an unwanted phase [48–50].

This work opens many possible research directions in the field of open quantum many-body systems. Our approach can be easily generalized to higher spatial dimensionality, thus allowing the study of this and similar models in dimensions $D > 2$. This could be exploited to study the fate of bistability in hyper-cubic lattices, as well as to investigate the effect of dissipators with different chiral structures, allowed in higher dimensional systems. In particular, the choice of chiral plaquettes with different geometries can reveal alternative, more intricated, dynamical patterns leading to bistability. A different direction corresponds to the comparison of the results obtained with CMF with other approximations such as the cumulant expansion [59]. The two approaches are indeed complementary [60], as CMF takes into account correlations of all order within a limited region, while the cumulant expansion treats correlations with no spatial restriction, but only up to a certain order. Therefore, comparing the two approaches can reveal novel features of the NEC model.

Our work highlights the connection between bistability and chiral kinetic constraints. These are often related to the emergence of Hilbert space fragmentation [16,17], where some parts of the Hilbert space are inaccessible to certain initial states. Investigating the possibility of a connection between bistability and fragmentation in the NEC or similar models represents a promising direction for future studies [19,61].

Finally, the NEC model can provide an interesting direction within the framework of measurement and feedback [62,63]. In this context, the dissipative processes characteristic of the NEC model could be implemented as a measurement on the plaquette followed by the action of the desired operator. This could provide a feasible experimental realization of the model, e. g. in dual-species Rydberg experiments [64], where our findings could be tested.

## Acknowledgments

We thank A. Nunnenkamp for insightful discussions. P. B. acknowledges support by the Austrian Science Fund (FWF) [Grant Agreement No. 10.55776/ESP9057324]. A. B. acknowledges financial support from the Provincia Autonoma di Trento and by the European Union — NextGeneration EU, within PRIN 2022, PNRR M4C2, Project TANQU 2022FLSPAJ [CUP B53D23005130006].

## A   Phase diagrams for all Hamiltonians

In Fig. 3 in the main text we reported the complete phase diagram of the NEC model under the coherent dynamics of the $H^{\text{PXP}}_{\bullet\bullet}$ Hamiltonian. For completeness, here we report the analogous phase diagrams obtained for the other models discussed.

In Fig. A1, we show the phase diagram obtained for $\ell = 3$ and $\Omega = 0.1$. As we mentioned in the main text, all the phase diagrams are qualitatively similar, sharing the same features. This suggests that the quantum fluctuations introduced by the coherent part of the dynamics change the behavior of the system only quantitatively, with respect to the underlying NEC

dissipative dynamics.

As we observed already in Fig. 5, the different models perturb in different ways the dissipative NEC phase diagram. In particular, we notice that $H_X$ is the most effective in destroying the ordered phase, reducing the critical amplitude $T_c$ at which the order parameter vanishes for all values of $h$. On the contrary, the PXP Hamiltonian shows a very stable phase diagram.

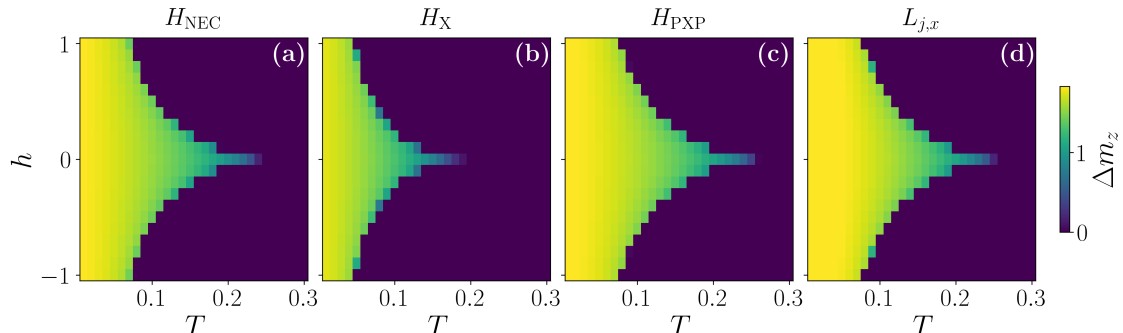

Figure A1: Comparison of the bistability phase diagrams at $\ell = 3$ and $\Omega = 0.1$ for the different models presented in the main text. The phase boundaries have the same qualitative shape, irrespective of the microscopic details, suggesting that this is a universal characteristic of the underlying chiral dissipative part. The quantitative shift agrees with the results presented in Fig. 5 in the main text.

We further provide additional details regarding the stability of the CMF ansatz in the different models, comparing the phase boundaries at $\ell = 2, 3$. In Fig. A2 (a) we show the critical value of the quantum fluctuations $\Omega_c$ for the Hamiltonian $H_X$, while in panel (b) we show the critical dissipation rate $\Gamma_c$ for the incoherent spin rotations implemented by the jump operators $L_{j,x}$. In both cases we compare the results obtained in the CMF with $\ell = 2$ (dashed lines) and $\ell = 3$ (solid lines) for different values of the bias. The comparison of the phase boundaries confirm the convergence of the CMF results at $\ell = 3$. Finally, in Fig. A2 (c) we compare the hysteresis cycle for $H^{\text{PXP}}$, showing a very good agreement between the different values of $\ell$.

# B Considerations on the dynamics of generic initial states

In the main text, we discussed the dynamics towards the steady-state in the hysteresis cycle, where the initial state corresponds to the steady-state of the previous bias instance. Here, we investigate how generic states evolve under the NEC rule in the cluster mean-field approximation.

First, we consider a generic bistable system, where there exist two different steady-states $\rho_{ss}^{(1)}$ and $\rho_{ss}^{(2)}$. Then, by linearity of the Lindlbad master equation, any linear combination of the two steady-states is a steady-state itself. However, in the cluster mean-field approximation, the Lindblad master equation is non-linear due to the expectation values decoupling the boundary terms.

We define $\mathcal{L}^{(1)} = \mathcal{L}^{\text{CMF}}(t)|_{\rho_{ss}^{(1)}}$ the Liouville super operator in the cluster mean-field approximation evaluated from $\rho_{ss}^{(1)}$, and analogously for the other steady-state. Then the equation of motion in the cluster mean-field approximation for $\rho = \alpha\rho_{ss}^{(1)} + \beta\rho_{ss}^{(2)}$ becomes

$$\dot{\rho} = (\alpha\mathcal{L}^{(1)} + \beta\mathcal{L}^{(2)})(\alpha\rho_{ss}^{(1)} + \beta\rho_{ss}^{(2)}) \tag{B1}$$

and the initial state is not necessarily a steady-state anymore due to the cross terms. Therefore, even states obtained by mixing the steady-states will eventually flow towards one of the two

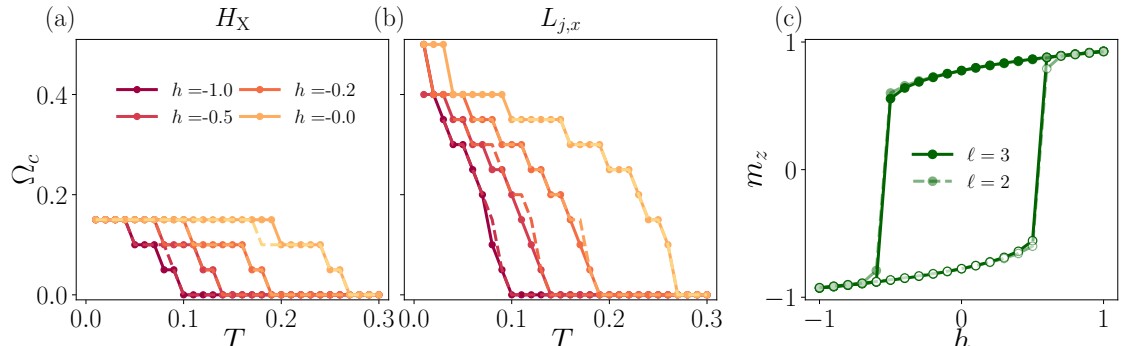

Figure A2: (a): Comparison of the phase boundaries for the Hamiltonian $H_X$ between the CMF ansatz at $\ell = 2$ (dashed lines) and $\ell = 3$ (solid lines). The results are insensitive to the variation of the cluster size, suggesting that the relevant correlations are correctly captured. (b): Similar comparison for the dissipative processes introduced by $L_{j,x}$. Also in this case the results show convergence of the CMF approach. (c): Comparison of the hysteresis cycle for the NEC Hamiltonian at $\Omega = T = 0.1$ shows very good agreement between the magnetization at $\ell = 3$ and at $\ell = 2$.

steady-states. This is in agreement with our analysis showing that generic states flow towards one of the two steady-states within the bistable phase.

In spite of this, in the thermodynamic limit, we expect the two steady-states to be orthogonal to one another, and therefore generic initial states will have large overlap only with one of the two. This would lead to their evolution towards a single steady-state, irrespective of the cluster mean-field approximation. This intuition is in agreement with our numerical simulations, showing stability with respect to the increase of the size of the plaquette.

## C  Details on the stability analysis

In the main text, we presented the stability analysis of the NEC model. Here, we give a detailed derivation of the equations used in the main text and we give the expression of the corner term.

In the stability analysis, we care about the fate of fluctuations on top of the steady state of the $n$-th cluster, $\delta\rho^{(n)}$. We thus write the cluster mean field equation for a state $\rho^{(n)} = \rho_{ss} + \delta\rho^{(n)}$

$$\dot{\rho}^{(n)} = \dot{\rho}_{ss} + \dot{\delta\rho}^{(n)} = \dot{\delta\rho}^{(n)} = -\imath[H_{\mathrm{CMF}}, \rho_{ss} + \delta\rho^{(n)}] + \mathcal{L}_{\mathrm{CMF}}[\rho_{ss} + \delta\rho^{(n)}]. \tag{C1}$$

One could be tempted to simply split the rhs of the equation into the steady state and fluctuation parts, but that would be wrong, since the cluster mean field equation is non-linear in the state, as it involves expectation values.

To make progress, we split the Hamiltonian and dissipators into cluster terms $H_{\mathcal{C}}$, $\mathcal{L}_{\mathcal{C}}$, acting linearly on the state $\rho^{(n)}$, and boundary terms, which instead involve expectation values. The latter will be split into $x$, $y$ and corner terms, corresponding to the right boundary, the top boundary and the top right corner of the cluster. Due to the structure of the plaquette, this site is special and involves expectation values evaluated on 2 different neighboring clusters. Therefore,

$$\dot{\delta\rho}^{(n)} = -\imath\Big([H_{\mathcal{C}}, \rho_{ss}] + [H_{\mathcal{C}}, \delta\rho^{(n)}] + [H_{\mathcal{B}_x}, \rho_{ss} + \delta\rho^{(n)}] + [H_{\mathcal{B}_y}, \rho_{ss} + \delta\rho^{(n)}][H_{\mathcal{B}_c}, \rho_{ss} + \delta\rho^{(n)}]\Big)$$
$$+ \mathcal{L}_{\mathcal{C}}[\rho_{ss}] + \mathcal{L}_{\mathcal{C}}[\delta\rho^{(n)}] + \mathcal{L}_{\mathcal{B}_x}[\rho_{ss} + \delta\rho^{(n)}] + \mathcal{L}_{\mathcal{B}_y}[\rho_{ss} + \delta\rho^{(n)}] + \mathcal{L}_{\mathcal{B}_c}[\rho_{ss} + \delta\rho^{(n)}].$$
$$\tag{C2}$$

Let us now analyze the boundary terms, starting with $\mathcal{B}_x$ and $\mathcal{B}_y$, which can be treated on the same footing. For the Hamiltonian part, we can write

$$
\begin{aligned}
[H_{\mathcal{B}_q}, \rho_{ss} + \delta\rho^{(n)}] &= \sum_{j\in\mathcal{B}_q} \mathrm{tr}[h_{j+e_q}(\rho_{ss}+\delta\rho^{(n)})][h_j, \rho_{ss}+\delta\rho^{(n)}] \\
&= \sum_{j\in\mathcal{B}_q} \Big(\mathrm{tr}[h_{j+e_q}\rho_{ss}] + \mathrm{tr}[h_{j+e_q}\delta\rho^{(n)}]\Big)\Big([h_j,\rho_{ss}] + [h_j,\delta\rho^{(n)}]\Big),
\end{aligned}
\tag{C3}
$$

where $h_j$ has support within the cluster, and $h_{j+e_q}$ has support on the neighboring cluster to the right $q = x$ or to the top $q = y$. For the dissipators, one has to be slightly more careful, as in the jump term they act on both sides of the density matrix. In our cluster mean field approach, the boundary jump operators are replaced by $\mathrm{tr}[l_{j+e_q}\rho]l_j$, thus

$$
\begin{aligned}
\mathcal{L}_{\mathcal{B}_q}[\rho_{ss}+\delta\rho^{(n)}] &= \sum_{j\in\mathcal{B}_q} \mathrm{tr}[l_{j+e_q}(\rho_{ss}+\delta\rho^{(n)})]l_j(\rho_{ss}+\delta\rho^{(n)})l_j^\dagger \mathrm{tr}[l_{j+e_q}^\dagger(\rho_{ss}+\delta\rho^{(n)})] \\
&\quad -\frac{1}{2}\mathrm{tr}[l_{j+e_q}^\dagger(\rho_{ss}+\delta\rho^{(n)})]\mathrm{tr}[l_{j+e_q}(\rho_{ss}+\delta\rho^{(n)})]\{l_j^\dagger l_j, \rho_{ss}+\delta\rho^{(n)}\} \\
&= \sum_{j\in\mathcal{B}_q} |\mathrm{tr}[l_{j+e_q}(\rho_{ss}+\delta\rho^{(n)})]|^2\Big[l_j\rho_{ss}l_j^\dagger + l_j\delta\rho^{(n)}l_j^\dagger - \frac{1}{2}\Big(\{l_j^\dagger l_j, \rho_{ss}\} + \{l_j^\dagger l_j, \delta\rho^{(n)}\}\Big)\Big].
\end{aligned}
\tag{C4}
$$

We notice that the jump term is of the third order in the state, thus differing from the commutator and anticommutator, which are second order.

We now analyze the corner boundary term, $\mathcal{B}_c$. In the top right corner of the plaquette, operators are evaluated on two different neighboring plaquettes, the one in the $y$ direction and the one in the $x$ direction. To account for these contributions, we write

$$
\begin{aligned}
[H_{\mathcal{B}_c}, \rho_{ss} + \delta\rho^{(n)}] &= \mathrm{tr}\Big[h_{c+e_x}(\rho_{ss}+\delta\rho^{(n)})\Big]\mathrm{tr}\Big[h_{c+e_y}(\rho_{ss}+\delta\rho^{(n)})\Big][h_c,\rho_{ss}+\delta\rho^{(n)}] \\
&= \Big[\big(\mathrm{tr}\big[h_{c+e_x}\rho_{ss}\big] + \mathrm{tr}\big[h_{c+e_x}\delta\rho^{(n)}\big]\big)\big(\mathrm{tr}\big[h_{c+e_y}\rho_{ss}\big] + \mathrm{tr}\big[h_{c+e_y}\delta\rho^{(n)}\big]\big)\Big]\big([h_c,\rho_{ss}] + [h_c,\delta\rho^{(n)}]\big)
\end{aligned}
\tag{C5}
$$

and for the dissipative part

$$
\begin{aligned}
\mathcal{L}_{\mathcal{B}_c}[\rho_{ss}+\delta\rho^{(n)}] &= \Big|\mathrm{tr}[l_{c+e_x}(\rho_{ss}+\delta\rho^{(n)})]\Big|^2\Big|\mathrm{tr}[l_{c+e_y}(\rho_{ss}+\delta\rho^{(n)})]\Big|^2\Big[l_c\rho_{ss}l_c^\dagger + l_c\delta\rho^{(n)}l_c^\dagger \\
&\quad -\frac{1}{2}\big(\{l_c^\dagger l_c, \rho_{ss}\} + \{l_c^\dagger l_c, \delta\rho^{(n)}\}\big)\Big].
\end{aligned}
\tag{C6}
$$

Due to the contribution of the two different neighboring clusters, the corner boundary term is a higher order functional of the density matrix, with respect to the ordinary boundaries.

Combining the results of the equations above, and plugging them into Eq. (C2) we then obtain Eq. (26) in the main text. We now group together the terms acting on the cluster with the ones where the expectation value is evaluated on the steady state, resulting in the cluster mean field super operator $\mathcal{M}_{\mathrm{CMF}}$

$$
\begin{aligned}
\mathcal{M}_{\mathrm{CMF}} &= -\imath[H_{\mathcal{C}}, \cdot] + \mathcal{L}_{\mathcal{C}} + \sum_{q=x,y}\sum_{j\in\mathcal{B}_q}\mathrm{tr}[h_{j+e_q},\rho_{ss}][h_j,\cdot] + |\mathrm{tr}[l_{j+e_q}\rho_{ss}]|^2\Big(l_j\cdot l_j^\dagger - \frac{1}{2}\{l_j^\dagger l_j,\cdot\}\Big) \\
&\quad -\imath\,\mathrm{tr}[h_{c+e_x}\rho_{ss}]\mathrm{tr}[h_{c+e_y}\rho_{ss}][h_c,\cdot] + |\mathrm{tr}[l_{c+e_x}\rho_{ss}]|^2|\mathrm{tr}[l_{c+e_y}\rho_{ss}]|^2\Big(l_c\cdot l_c^\dagger - \frac{1}{2}\{l_c^\dagger l_c,\cdot\}\Big).
\end{aligned}
\tag{C7}
$$

By definition, $\mathcal{M}_{\mathrm{CMF}}[\rho_{ss}] = 0$. Neglecting non-linear terms in the fluctuations and noting that in our case the operators $l_{j+e_q}$ are always Hermitian, we obtain

$$
\dot{\delta\rho}^{(n)} = \mathcal{M}_{\mathrm{CMF}}[\delta\rho^{(n)}] + \sum_{q=x,y}\sum_{j\in\mathcal{B}_q}[h_j,\rho_{ss}]\mathrm{tr}[h_{j+e_q}\delta\rho^{(n)}] + 2\mathrm{tr}[l_{j+e_q}\rho_{ss}]\left(l_j\rho_{ss}l_j^\dagger - \frac{1}{2}\{l_j^\dagger l_j,\rho_{ss}\}\right)\mathrm{tr}[l_{j+e_q}\delta\rho^{(n)}]
$$

$$
-\iota[h_c,\rho_{ss}]\left(\mathrm{tr}[h_{c+e_x}\rho_{ss}]\mathrm{tr}[h_{c+e_y}\delta\rho^{(n)}] + \mathrm{tr}[h_{c+e_y}\rho_{ss}]\mathrm{tr}[h_{c+e_x}\delta\rho^{(n)}]\right)
$$

$$
+ 2\mathrm{tr}[l_{c+e_x}\rho_{ss}]\mathrm{tr}[l_{c+e_y}\rho_{ss}]\left(l_c\rho_{ss}l_c^\dagger - \frac{1}{2}\{l_c^\dagger l_c,\rho_{ss}\}\right)\left(\mathrm{tr}[l_{c+e_x}\rho_{ss}]\mathrm{tr}[l_{c+e_y}\delta\rho^{(n)}] + \mathrm{tr}[l_{c+e_y}\rho_{ss}]\mathrm{tr}[l_{c+e_x}\delta\rho^{(n)}]\right)
$$

$$\tag{C8}$$

corresponding to Eq. (27) in the main text.

Following the same steps discussed in the main text, we can write the equation above as a super operator acting on the fluctuation $\delta\rho^{(n)} = \sum_{\mathbf{k}}e^{\iota\mathbf{k}\cdot\mathbf{r}_n}\delta\rho_{\mathbf{k}}$. Upon vectorization, we then obtain

$$
\partial_t\|\delta\rho_{\mathbf{k}}\rangle\rangle = \left[\mathbb{M}_{\mathrm{CMF}} + \sum_{\substack{q=x,y\\j\in\mathcal{B}_q}}e^{\iota k_q}\left(-\iota\|\rho_{h_j}\rangle\rangle\langle\langle h_{j+e_q}\| + 2\Re\left[\mathrm{tr}\left[l_{j+e_q}\rho_{ss}\right]\right]\|\rho_{\mathcal{L}_j}\rangle\rangle\langle\langle l_{j+e_q}\|\right)\right.
$$

$$
-\iota\left(\mathrm{tr}[h_{c+e_x}\rho_{ss}]\|\rho_{h_c}\rangle\rangle\langle\langle h_{c+e_y}\| + \mathrm{tr}[h_{c+e_y}\rho_{ss}]\|\rho_{h_c}\rangle\rangle\langle\langle h_{c+e_x}\|\right)
$$

$$
\left.+ 2\mathrm{tr}[l_{c+e_x}\rho_{ss}]\mathrm{tr}[l_{c+e_y}\rho_{ss}]\left(\mathrm{tr}[l_{c+e_x}\rho_{ss}]\|\rho_{\mathcal{L}_c}\rangle\rangle\langle\langle l_{c+e_y}\| + \mathrm{tr}[l_{c+e_y}\rho_{ss}]\|\rho_{\mathcal{L}_c}\rangle\rangle\langle\langle l_{c+e_x}\|\right)\right]\|\delta\rho_{\mathbf{k}}\rangle\rangle,
$$

$$\tag{C9}$$

corresponding to Eq. (28) in the main text.

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
