# Peer review of "Dissipative quantum North-East-Center model: steady-state phase diagram, universality and nonergodic dynamics"

_SciPost Physics_

## Round 1 · Referee Report · Anonymous (Referee 3) · 2025-12-2

Report

The authors have addressed the main point I asked about well, both in the reply and in the paper itself. I recommend that this be published in SciPost Physics.

Recommendation

Publish (meets expectations and criteria for this Journal)

---

## Round 1 · Referee Report · Anonymous (Referee 1) · 2025-12-4

Strengths

1- careful analysis of a dissipative quantum system displaying bistability.

Weaknesses

1- special model 2- cluster mean-field analysis neglecting fluctuations between clusters

Report

The manuscript has been improved by accounting for the Referees' comments.
In this respect, it clearly has become publishable. But I see the progress made in this work with respect to literature more incremental than seminal. In view of this and the quite special model for which no experiments are envisaged I do not see one of the four acceptance criteria (expectations) fulfilled. So I recommend publication in SciPost Physics Core.

Recommendation

Accept in alternative Journal (see Report)

---

## Round 1 · Referee Report · Anonymous (Referee 2) · 2025-12-8

Report

The authors have addressed all the comments and clarified the manuscript. I especially appreciate the expansion of Appendix C.
I support the publication in SciPost Physics.

Requested changes

Even though the authors now define what is meant by "chiral plaquette symmetry", it seems to me that this is rather an abuse of the term "symmetry", since what is meant is not a symmetry (i.e. an invariance under some transformation) but an asymmetric geometry of the plaquette interactions. As such, saying that the Hamiltonian in (10) has no plaquette symmetry seems paradoxical for, strictly speaking, that translationally invariant Hamiltonian has more symmetry than the plaquette terms. Even though within the context of the paper one can understand what is meant, I would suggest to talk about "plaquette chiral geometry" (or similar) rather than symmetry.

Recommendation

Publish (meets expectations and criteria for this Journal)

  • validity: -
  • significance: -
  • originality: -
  • clarity: -
  • formatting: -
  • grammar: -

Author:  Pietro Brighi  on 2025-12-09  [id 6124]

(in reply to Report 3 on 2025-12-08)

We thank the Referee for this suggestion, which will be implemented in the updated version of the manuscript.

---

## Round 1 · Author Response

Dear Editor,
We carefully read the referee reports together with your assessment of the Manuscript.
We are happy to see that two out of three reports consider the paper suitable for SciPost Physics after major and minor revisions, respectively. However, the first referee feels that the paper is more suitable for SciPost Physics Core.

We argue that the paper is well-suited for SciPost Physics since it presents a significant advance on an important class of dissipative spin lattices featuring bistability and collective behaviours. We feel that in the previous version of the manuscript we did not stress enough how we go beyond the existing literature, specifically with respect to the Ref.[45].

To this aim in the revised version we commented clearly about this point in the Introduction. Also, we detailed our claims in the reply to Referee three. Here we report the main points:

We go systematically beyond the variational ansatz proposed in Ref.[45] by means of the cluster mean field approach (extensively used in the literature for 2D dissipative spin lattices).
Equipped with this tool we determine the structure of the phase diagram and quantitatively locate its boundaries. In the revised version of the Manuscript we also included simulation for a 4X4 plaquette, that is numerically the state of the art.
We identify the mechanisms responsible for the bubble reabsorptions and we propose an equation for their dynamics that we validate numerically (developing a non-translationally invariant version of the cluster mean-field ansatz).

We hope that the revised version of the Manuscript can be now considered suitable for publication in SciPost Physics.

Best regards

---

## Round 1 · List of Changes

List of the main changes :

1) In the Introduction we now briefly describe the important advancements with respect to Ref.[45]. 2) Following the referee suggestions we included the analysis of a larger plaquette made of 4X4 sites. 3) We clarified the meaning of the stability analysis, adding a more detailed connection with the cluster mean-field analysis and extending the Appendix with derivation of the equations in the main text. 4) We included all changes requested by the referees regarding figures formatting and other smaller issues.

---

## Editorial Decision

accepted_in_target_journal